# Host Tau Genotype Specifically Designs and Regulates Tau Seeding and Spreading and Host Tau Transformation Following Intrahippocampal Injection of Identical Tau AD Inoculum

**DOI:** 10.3390/ijms23020718

**Published:** 2022-01-10

**Authors:** Pol Andrés-Benito, Margarita Carmona, Mónica Jordán, Joaquín Fernández-Irigoyen, Enrique Santamaría, José Antoni del Rio, Isidro Ferrer

**Affiliations:** 1Neuropathology Group, Institute of Biomedical Research, IDIBELL, L’Hospitalet de Llobregat, 08907 Barcelona, Spain; pandres@idibell.cat (P.A.-B.); mcarmona@idibell.cat (M.C.); 2CIBERNED (Network Centre of Biomedical Research of Neurodegenerative Diseases), Institute of Health Carlos III, L’Hospitalet de Llobregat, 08907 Barcelona, Spain; mjordanpirla@gmail.com (M.J.); jadelrio@ibecbarcelona.eu (J.A.d.R.); 3Clinical Neuroproteomics Unit, Proteomics Platform, Proteored-ISCIII, Navarrabiomed, Complejo Hospitalario de Navarra (CHN), Universidad Pública de Navarra (UPNA), diSNA, 31008 Pamplona, Spain; joaquin.fernandez.irigoyen@navarra.es (J.F.-I.); enrique.santamaria.martinez@navarra.es (E.S.); 4Molecular and Cellular Neurobiotechnology, Institute of Bioengineering of Catalonia (IBEC), Science Park Barcelona (PCB), Barcelona Institute for Science and Technology, 08028 Barcelona, Spain; 5Department of Cell Biology, Physiology and Immunology, Faculty of Biology, University of Barcelona, 08028 Barcelona, Spain; 6Department of Pathology and Experimental Therapeutics, University of Barcelona, L’Hospitalet de Llobregat, 08907 Barcelona, Spain

**Keywords:** host tau, seeding and spreading, 3Rtau and 4Rtau, hTau, Alzheimer’s disease, tauopathies

## Abstract

Several studies have demonstrated the different characteristics of tau seeding and spreading following intracerebral inoculation in murine models of tau-enriched fractions of brain homogenates from AD and other tauopathies. The present study is centered on the importance of host tau in tau seeding and the molecular changes associated with the transformation of host tau into abnormal tau. The brains of three adult murine genotypes expressing different forms of tau—WT (murine 4Rtau), hTau (homozygous transgenic mice knock-out for murine tau protein and heterozygous expressing human forms of 3Rtau and 4Rtau proteins), and mtWT (homozygous transgenic mice knock-out for murine tau protein)—were analyzed following unilateral hippocampal inoculation of sarkosyl-insoluble tau fractions from the same AD and control cases. The present study reveals that (a) host tau is mandatory for tau seeding and spreading following tau inoculation from sarkosyl-insoluble fractions obtained from AD brains; (b) tau seeding does not occur following intracerebral inoculation of sarkosyl-insoluble fractions from controls; (c) tau seeding and spreading are characterized by variable genotype-dependent tau phosphorylation and tau nitration, MAP2 phosphorylation, and variable activation of kinases that co-localize with abnormal tau deposits; (d) transformation of host tau into abnormal tau is an active process associated with the activation of specific kinases; (e) tau seeding is accompanied by modifications in tau splicing, resulting in the expression of new 3Rtau and 4Rtau isoforms, thus indicating that inoculated tau seeds have the capacity to model exon 10 splicing of the host *mapt* or *MAPT* with a genotype-dependent pattern; (e) selective regional and cellular vulnerabilities, and different molecular compositions of the deposits, are dependent on the host tau of mice injected with identical AD tau inocula.

## 1. Introduction

Neurodegenerative diseases with abnormal protein aggregates are characterized neuropathologically by the deposition of protein aggregates resulting mainly from aberrant post-translational modifications of primary constitutive elements of the nervous system. The term tauopathy covers heterogeneous diseases having in common the neuronal and glial deposition of abnormally phosphorylated species of tau protein, usually accompanied by other post-translational modifications in tau [1,2,3].

Selective neuronal and glial vulnerability and progression are determinants in neurodegenerative diseases with abnormal protein aggregates [4,5]. Several in vivo studies demonstrate the capacity for seeding and spreading of tau following intracerebral inoculation of transgenic mice expressing 4R human tau or mutant human tau. Tau seeding and spreading is produced following the intracerebral inoculation of synthetic tau fibrils [6,7] or inoculation of fibrillar-enriched fractions from human and mouse brain homogenates of tauopathies, including Alzheimer’s disease (AD), tangle-only dementia progressive supranuclear palsy (PSP), corticobasal degeneration (CBD), and argyrophilic grain disease (AGD), as well as tau transgenic mice in various transgenic mouse models [8,9,10,11,12]. Tau seeding, and tau seeding and spreading, also occurs following intracerebral inoculation of similar tau aggregates in wild type mice (WT) and in transgenic WT mice [9,13,14,15]. In addition to neurons, deposits also occur in glial cells, and the morphology of glial inclusions appears to mimic the glial aggregates of the corresponding human tauopathies in inoculated transgenic mice expressing human tau or mutant human tau, and in WT mice [9,10,11,12,14,15,16].

It is widely accepted that tau strains are behind the different phenotypes and progression of human tauopathies [5,17,18,19,20]. Abnormal tau may also have different effects on the host; for example, P301L seeds are uniquely modified by post-translational modifications within the microtubule-binding region resulting from histone deacetylase 6 (HDAC6) inhibition and tau auto-acetylation; HDAC6 inhibition results in accelerated tau aggregation [21].

However, our previous studies of tau seeding and spreading in WT mice show similar cellular vulnerability despite the origins of the inocula. Inoculation of sarkosyl-insoluble fractions of brain homogenates from human tauopathies, including AD, primary age-related tauopathy (PART), aging-related tau astrogliopathy (ARTAG), PSP, AGD, Pick’s disease (PiD), frontotemporal lobar degeneration linked to MAPT P301L mutation, and globular glial tauopathy (GGT) reproduces a tauopathy, but inoculated mice do not express the markers of specific tauopathies [22,23,24,25,26]. Pick bodies, thorn-shaped astrocytes, tufted astrocytes, astrocytic plaques, and glial globular inclusions are not observed, but coiled bodies in oligodendrocytes are constant even following inoculation of AD and PART homogenates, representing diseases in which no tau deposits occur in oligodendroglia [24]. Conversely, intracerebral inoculation of brain homogenates of pure ARTAG cases, which do not have tau deposits in neurons, induces neuronal tau seeding and spreading in addition to oligodendroglial inclusions [22]. These observations suggest that not only the origin of the donor tau but also the characteristics of the host tau are relevant in tau seeding and spreading [26], and that oligodendrocytes in WT mice are particularly vulnerable to human tau inoculation [23].

Another relevant observation in our inoculation paradigm using WT mice is the presence of 3Rtau and 4Rtau deposits in neurons and oligodendrocytes in adult mice which express only 4Rtau in most brain regions under physiological conditions. Thus, inoculated homogenates, regardless of their primary tau composition, be it 3Rtau plus 4Rtau (from AD and PART cases), only 4Rtau (obtained from pure ARTAG, PSP, GGT, and FTLD-P301L cases), or 3Rtau (obtained from PiD cases), produce 3Rtau and 4Rtau deposits in WT mice [22,23,24,25,26]. This suggests a shift in the modulation of exon 10 splicing following tau inoculation obtained from distinct tauopathies.

The present study analyzes the characteristics of tau seeding and spreading following intrahippocampal inoculation of insoluble tau (sarkosyl-insoluble fractions) from the same AD case to three murine models expressing different forms of tau: WT (murine 4Rtau), hTau (homozygous transgenic mice knock-out for murine tau protein and heterozygous expressing human forms of 3Rtau and 4Rtau proteins) [27,28], and mtWT (homozygous transgenic mice knock-out for murine tau protein). Our endeavor was to gain understanding of the different regulation and modelling of tau seeding and spreading depending on the characteristics of host tau, and the molecular changes associated with the transformation of host tau following inoculation of brain insoluble fractions from the same donor.

## 2. Results

### 2.1. WT Mice, Mutant Wild-Type KO for Murine Tau (mtWT), and Tau Transgenic Mice Expressing Human Tau (hTau)

The telencephalon of WT mice was not stained with anti-3Rtau antibodies with the exception of polymorphic cells of the inner region of the dentate gyrus, and a few neurons in the entorhinal cortex. The neuropil of the telencephalon showed weak and diffuse 4Rtau immunoreactivity. Anti-PHF1 antibodies revealed moderate immunoreactivity in the neuropil but not in the cell bodies, whereas MC1 immunoreactivity was negative.

The brain of mtWT mice was not stained with anti-tau antibodies.

hTau transgenic mice showed discrete PHF1 and MC1 immunoreactivity in the cytoplasm of scattered neurons in the entorhinal cortex, hippocampus, dentate gyrus, and cerebral cortex. Positive neurons were rare in the thalamus, and absent in the striatum. Weak PHF1 immunoreactivity was also found in the neuropil, whereas MC1-immunoreactive deposits were restricted to the cytoplasm of a subpopulation of neurons. Diffuse 3Rtau was more marked than 4Rtau immunoreactivity in hTau mice. In the neuropil, 3Rtau predominated, whereas 4Rtau decorated the cytoplasm of neurons. Similar staining was seen in mice aged 6 and 9 months with no evidence of tau pathology progression. Images of the CA1 region of the hippocampus and dentate gyrus in the three groups of mice are shown in Figure 1 for comparison with the corresponding regions in inoculated mice (see later).

Densitometric studies further stressed the differences in the expression of 3Rtau and 4Rtau in the CA1 region ad dentate gyrus in the three groups of mice (Figure 1 graphs). As expected 4Rtau is the predominant form in WT mice, but the presence of 3Rtau-immunoreactive neurons in the inner region of the DG is also represented by a 3Rtau signal in the DG of WT mice. In contrast, 3Rtau is more abundant in the CA1 region and DG in hTau. mtWT mice do not show 3Rtau and 4Rtau expression.

Neurons and glial cells in WT mice, mtWT, and hTau transgenic mice were not stained with anti-tau antibodies tau-P Ser422, tau AT8-P Ser202/Thr205 (excepting non-specific staining of the nuclei of neurons in WT mice), tau C-3, and tau-N Tyr29Tau at the assessed time points (6 and 9 months) (Table 1).

Neurons in WT, mtWT and hTau mice were not stained with antibodies MAP2-P Thr1620/1623, NFL-P Ser473, casein kinase δ, p38-P Thr180/tyr182, SAPK/JNK-P Thr183/Thr185, and SRC-P Tyr416. Ubiquitin deposits were absent. We did not detect abnormal structures with anti-eIF2α-P Ser51, IRE-P Ser274 GSK-3β-P Ser9, LAMP1, and LC3 antibodies. AKT-P Ser473, PKAα/β-P Tyr197, SRPK1, and SRPK2-P Ser497 antibodies did not show abnormal deposits in the three groups of mice (Table 1).

### 2.2. Western Blotting of Total Brain Homogenates in WT, hTau, and Mutant Wild-Type KO for Murine Tau (mtWt)

Total brain homogenates were used to identify tau profiles in the three groups of mice at the age of 9 months. Gel electrophoresis and Western blotting were processed in parallel. The antibody Tau 5 that identifies total tau showed weak bands of about 68 kDa and 64 kDa. Strong bands of about 68 kDa, 64 kDa, and 60–50 kDa, together with smears of lower molecular weight, were identified in hTau mice. In WT mice, 3Rtau was negative, but two bands of about 64 kDa and 60–50 kDa and a weak band of slightly higher molecular weight were detected in hTau mice. A weak 4Rtau-immunoreactive band of about 60–50 kDa was present in WT mice, as were two bands of 68 kDa and 64 kDa in hTau mice. The immunoreactivity of 3Rtau was higher when compared with 4Rtau in hTau transgenic animals. Phosphorylated tau at Thr181 was absent in WT mice, but several bands of 68 kDa and 64 kDa were identified in hTau. PHF1 antibodies recognized a unique band of about 64 kDa in WT mice. We do not know the origin of this band, but it matches the diffuse PHF1 immunostaining of the neuropil in WT mice as revealed by immunohistochemistry. The pattern of PHF1 in hTau transgenic was characterized by three bands of 68 kDa, 64 kDa, and 60–50 kDa. Finally, MC1 antibodies recognized several bands of 68 kDa, 64 kDa, and 60–50 kDa in hTau, but not in WT mice. Although MC1 is an antibody raised against abnormal conformation of tau, and the conformational structure might be modified by the process of electrophoresis, the presence of the band profile in hTau mice indicates structural preservation. Finally, as expected, Western blots were negative in mtWT mice to all forms of tau (Figure 2).

### 2.3. Characteristics of Sarkosyl-Insoluble Fractions Used for Inoculation

Sarkosyl-insoluble fractions of AD brain homogenates blotted with anti-tauSer422 were characterized by three bands of 68 kDa, 64 kDa, and 60 kDa, a weak upper band of 73 kDa, and several lower bands of fragmented tau between 50 and 25 kDa. Lower bands stained with anti-tauSer422 indicated truncated tau at the C-terminal. Sarkosyl-insoluble fractions of the control case blotted with the same antibody were negative (Figure 3A).

### 2.4. Inoculation of Mice with Sarkosyl-Insoluble Fractions from AD and Control; mtWT Mice, and Characteristics of Tau Seeding in WT and hTau Genotypes

Inoculated mice showed local astrocytic gliosis and variable numbers of phagocytes filled with hemosiderin and clear vacuoles along the course of the needle and at the site of the injection independently of the genotype (Figure 4A). These changes were interpreted as the result of non-specific traumatic injury of the intracerebral injection. Areas distant from the injection site, including the contralateral dentate gyrus, did not show increased numbers of astrocytes or microglia.

mtWT mice killed at three and six months after inoculation of sarkosyl-insoluble fractions from AD did not show any tau deposition. This observation is relevant as it demonstrates that inoculated tau does not freely circulate and is not stored in the host brain. Inoculated tau completely disappears at some time before three months of survival.

WT mice unilaterally inoculated with AD sarkosyl-insoluble fractions in the hippocampus gyrus at six months and killed at nine months (three months of survival) showed abnormal tau deposition, as revealed with the antibody AT8, in neurons of the dentate gyrus, cerebral cortex near the site of inoculation, CA1 region of the hippocampus, and threads and glial cells in the fimbria and ipsilateral corpus callosum (Figure 4B–E). Deposits in glial cells in the corpus callosum and fimbria were better visualized at higher magnification (Figure 4 F–H). Abnormal deposits were stained with anti-4Rtau (Figure 4I, J), and 3Rtau antibodies (Figure 4K,L). Abnormal deposits were also strongly immunostained with anti-MAP2-P Thr1620/1623 antibodies (Figure 4M–P). Tau deposits were not stained with antibodies Tau-P Ser422, PHF1, Tau-N Tyr29, MC1, MAP2-P Thr1620/1623, or casein kinase δ (data not shown). Expression of active (phosphorylated) kinases p38-P and SAPK/JNK-P was observed in phospho-tau deposits in inoculated WT mice, as reported elsewhere [22,23,24,25,26]. However, GSK-3βP Ser9, AKT-P Ser473, and PKA α/β-P Tyr197 immunoreactivity was absent (images not shown).

hTau mice unilaterally inoculated in the hippocampus with AD sarkosyl-insoluble fractions at six months and killed at nine months (three months of survival) showed abnormal tau deposition mainly in the CA1 region of the hippocampus and dentate gyrus, as seen with the antibody AT8 (Figure 5A,D). The size of neuronal deposits was greater and the morphology flame-shaped in hTau mice when compared with tau deposits in inoculated WT mice. Deposits also occurred in the hilus, cerebral cortex near the inoculation site, corpus callosum, and fimbria. Tau deposits were localized in neurons, threads, and granules. Deposits contained 4Rtau (Figure 5B,E), but 3Rtau was more abundant and higher in density, particularly in the CA1 region (Figure 5C,F).

Curiously, 3Rtau and 4Rtau deposits were localized in the cytoplasm of neurons which was in contrast with the predominant localization of 3Rtau in the neuropil in non-inoculated hTau mice (see Figure 1 for comparison). Tau deposits were stained with antibodies Tau-P Ser422, PHF1, Tau-N Tyr29, and, rarely, MC1 (Figure 5G–J), and they also contained MAP2-P Thr1620/1623 and casein kinase δ (Figure 5K,L). In addition, small cytoplasmic granules immunoreactive with anti-GSK-3βP Ser9, AKT-P Ser473, and PKA α/β-P Tyr197 antibodies were present in neuronal populations with abnormal tau deposits (Figure 5M–O). Cytoplasmic granules in tau-containing neurons were also stained with p38-P Thr180/Tyr182 and SAPK/JNK-P Thr183/Thr185 antibodies (see below in double-labelling immunofluorescence and confocal microscopy).

Differences in the intensity of the densitometric signal between 3Rtau and 4Rtau in the CA1 and DG of inoculated hTau mice are shown in the graph of Figure 5. Immunoreactivity of 3Rtau is significantly higher than 4Rtau immunoreactivity in the CA1 region. This pattern was not reproduced in the DG due to individual variations.

A marked difference between inoculated WT and hTau was the deposition of abnormal tau in numerous oligodendrocytes in the corpus callosum and fimbria, in addition to positive threads in inoculated WT mice. Positive threads were common in the plexiform layers of the hippocampus and to a lesser degree in the ipsilateral corpus callosum, but phospho-tau-containing oligodendrocytes were very rare in inoculated hTau mice. Astrocytes did not show tau deposits in WT and hTau inoculated mice.

The alterations were similar in mice inoculated at the age of three months and killed at nine months (survival time six months) excepting for tau spreading to the contralateral hippocampus in inoculated WT mice, as previously reported [23].

Neuronal and glial deposits in inoculated WT and hTau mice were devoid of reticulum stress markers eIF2α-P Ser51, IRE-P Ser274, and markers of autophagy LC3 and LAMP1 (data not shown).

Tau deposits were not detected in animals inoculated with homogenates from control brains. No tau deposits were seen in mice inoculated with vehicle alone.

The distribution of tau deposits in WT and hTau unilaterally inoculated in the hippocampus with the same inocula of sarkosyl-insoluble fraction of AD is shown in Figure 6. The involvement of the CA1 region and dentate gyrus was greater in inoculated hTau mice when compared with WT mice. Tau deposits in the stratum radiatum and stratum oriens were also more abundant in hTau than in WT mice at the same survival time. In contrast, tau deposits in the corpus callosum and fimbria were more marked in WT mice. For comparative purposes, the same figure illustrates the absence of tau deposits in mtWT inoculated with sarkosyl-insoluble fractions of AD, and the absence of tau deposits in WT and hTau mice inoculated with sarkosyl-insoluble fractions from a control case.

Double-labelling immunofluorescence and confocal microscopy to p38-P Thr180/Tyr182 and AT8, and SAPK/JNK-P Thr183/Thr185 and AT8 in inoculated hTau showed the co-localization of these active tau kinases and hyper-phosphorylated tau deposition (Figure 7A,B). Double-labelling immunofluorescence and confocal microscopy also showed co-localization of PKAα/β-P Tyr197-immunoreactive granules with phospho-tau deposits (Figure 7C).

Finally, we assessed the expression of selected molecules linked to exon 10 splicing. SR protein kinases SRPK1 and SRPK2-P Ser497 were expressed in the cytoplasm of neurons in non-inoculated and inoculated mice regardless of the genotype. Double-labeling immunofluorescence and confocal microscopy to AT8 and SRPK2-P Ser497 further revealed no differences between neurons without and with abnormal tau deposits in inoculated hTau mice. Figure 7D shows co-localization of SRPK2-P Ser497 and AT8 in the CA1 region of inoculated hTau mice.

Curiously, reactive astrocytes at the site of injection showed strong SRPK1 immunoreactivity in the cytoplasm and cell processes (data not shown). As a differentiating point between WT and hTau inoculated mice, punctate cytoplasmic CLK1 immunoreactivity was only observed in neurons of the CA1 region in inoculated hTau mice but not in inoculated WT mice (Figure 7E,F).

Observations of tau deposits in neurons are summarized semi-quantitatively in Table 1. 

## 3. Discussion

Several studies have demonstrated the capacity for seeding and spreading of tau-enriched fractions of brain homogenates from AD and other human and mouse tauopathies following intracerebral inoculation into transgenic mice bearing human tau or mutant human tau [8,9,11,12,16] and into WT and tgWT mice [9,13,14,15,16,22,23,24,25,26]. In addition to neurons, deposits occur in glial cells, and the morphology of glial inclusions appears to mimic the glial aggregates of the corresponding human tauopathies in inoculated transgenic mice expressing human tau or mutant human tau [10,11,12,13,14,15,16]. However, our previous studies showed differences between the phenotype of the original tauopathy and the resultant tauopathy following intracerebral inoculation of tau-enriched homogenates into WT mice [22,23,24,25,26]. In our experiments using WT mice, tau deposits in neurons resembled pre-tangles rather than tangles; oligodendrocytes contained tau deposits despite the tauopathy; and tau deposits were almost entirely absent in astrocytes excepting the first stages following tau ARTAG. We speculated that host tau, as does inoculated tau, plays a role in the capacity for tau seeding and spreading [26]. To assess this hypothesis, here we used three genotypes of mice carrying different forms of tau: adult WT mice expressing murine 4Rtau, hTau expressing human 3Rtau and 4Rtau in a KO-murine tau background, and mtWT mice that do not express murine and human tau. In our previous studies, we also speculated about the possibility that the transformation of host tau into abnormal tau is an active process associated with the activation of selected kinases [22,23,24,25,26]. This point lies beyond the mechanisms of tau extrusion and tau uptake linked to tau transmission already discussed in several publications [17,18,19].

hTau transgenic mice express high levels of human 3Rtau and low levels of 4Rtau in Western blots of brain homogenates. Phosphorylated tau deposition occurs in subpopulations of neurons in the cerebral cortex and hippocampus. Abnormal deposits are identified with the antibodies PHF1 and MC1, but tau deposits are not stained with AT8, tau-P Ser422, and Tau C-3 antibodies [27,28]. There is no apparent activation of tau kinases p38, SAPK/JNK, GSK-3β Ser9, and SRC kinases co-localizing with abnormal tau deposits in hTau mice. Casein kinase δ immunoreactivity and markers of granulovacuolar degeneration linked to endoplasmic reticulum stress and autophagy are negative in hTau. As expected, WT mice do not have tauopathy, and mtWT mice do not express any type of tau using the same battery of anti-tau antibodies.

We inoculated sarkosyl-insoluble fractions from the same AD case to randomize donor tau, and injected the same amount of the inoculum unilaterally into the same region of the hippocampus in every mouse. The band pattern of phospho-tau of AD in Western blots is characterized by a combination of 3Rtau and 4Rtau phosphorylated bands and by the presence of truncated forms, whereas no bands of phosphor-tau are identified in controls at the same exposure time. (Phospho)proteomics reveals enrichment in tau phosphorylated at different sites in AD. Phosphorylated tau is also present in control homogenates. However, phosphorylated tau species in controls do not form aggregates visualized in Western blots.

No deposits develop in mtWT mice. This is relevant as it indicates that host tau is essential for tau spreading in inoculated mice. Moreover, the latter experiment shows that the pathology of tau progression in inoculated WT and hTau mice is not due to the passive extension of the inocula.

In inoculated WT and hTau mice, differences in the regional vulnerability to tau seeding and spreading depend on the genotype. The morphology of neuronal tau deposits in inoculated WT are reminiscent of pre-tangles, but neuronal tau aggregates in inoculated hTau mice resemble neurofibrillary tangles. The involvement of the CA1 region is greater in hTau mice when compared with WT mice. Tau deposits in the stratum radiatum and stratum oriens are more abundant in hTau than in WT. In contrast, phospho-tau-containing threads in the corpus callosum and fimbria, and abnormal tau in oligodendrocytes, are marked in WT, but reduced and almost absent when dealing with oligodendrocytes, in hTau mice. It is worth noting that oligodendrocytes in normal conditions produce and express tau protein [30,31]. Abnormal tau deposition in oligodendrocytes occurs constantly in our paradigms following inoculation of tau from different tauopathies including AD, PART, AGD, PSP, ARTAG, and FTLD-P301L in WT mice [22,23,24,25,26]. The present observations show that vulnerability of oligodendrocytes is likely dependent on the host tau genotype.

The present results demonstrate that inoculation of abnormal tau triggers an active process of post-translational modifications of various proteins, including tau, in vulnerable cells, which is partially dependent on the host genotype. Tau deposits in inoculated WT and hTau are stained with antibodies AT8 and Tau-P Ser422, and this is accompanied by abnormal deposits of MAP2-P Thr1620/1623. Active phosphorylation of retrieved tau and MAP2 from the host following AD-tau inoculation is supported by the co-localization of active kinases p38 and SAPK-JNK with phospho-tau in inoculated WT and hTau mice. Moreover, abnormal protein deposits aggregates in inoculated hTau mice, but not in WT are stained with PHF1 and MC1 antibodies, contain tau-N Tyr29, and co-localize with cytoplasmic granules immunoreactive with CK1-δ, CLK1, GSK-3β-P Ser9, and PKAα/β-P Tyr197 antibodies.

Previous studies have shown the activation of the unfolded protein response in pre-tangle neurons, and the induction of granulovacuolar degeneration bodies (GVBs) following intracerebral inoculation of tau fibrils [32,33]. Seeding of tau pathology in vivo using synthetic K18 tau P301L PHFs seeds into the hippocampus of tau P301L Tg and induces granulovacuolar degeneration. Similarly, brain lysates of tau P301S Tg mice with filamentous tau injected into the hippocampus and overlaying cerebral cortex of WT tau ALZ17 mice, and injection with human post-mortem brain lysates from donors with confirmed neuropathology of AD and primary tauopathy induces GVBs immunoreactive for CK1δ and pPERK [33]. hTau mice, but not WT mice, inoculated with sarkosyl-insoluble fractions of AD homogenates show cytoplasmic granules in neurons with cytoplasmic tau deposits; these granules are immunoreactive with anti CK1-δ, CLK1, GSK-3β-PSer9, p38-PThr 180/Tyr182, PKA α/β Tyr197, and SAPK/JNK-P Thr183/Thr185 antibodies. However, abnormal deposits in neurons and glial cells are not stained with anti-eIF2α-P Ser51, IRE-P Ser274, LC3, and LAMP1 antibodies in our models, suggesting that they do not present many principal components of granulovacuolar degeneration (GVD) [34,35].

Tau deposits in WT and hTau inoculated with sarkosyl-insoluble fractions are composed of 4Rtau and 3Rtau. This is a curious feature as 4Rtau is the predominant isoform in adult WT mice (excepting a few neurons in the inner region of the DG in WT mice). Regarding hTau mice, 3Rtau is more abundant than 4Rtau in inoculated mice, but marked induction of 4Rtau occurs as well. A shift in the expression of 4Rtau to 3Rtau has been reported in oligodendrocytes following middle cerebral occlusion in adult rats and mice, thus suggesting exon 10 splicing modulation in under determinate conditions not limited to those occurring in the developing brain [36].

Several factors may contribute to the differences in tau deposits between WT and hTau mice intracerebrally injected with the same inocula. One of them is the different composition of murine and human tau at several sites including the amino- and carboxy-terminals and the capacity for binding to other substrates [37]. This is equivalent to the concept of the species barrier in prion diseases. Another is the modulation of tau splicing in the two species. A shift from fetal to adult tau isoform expression occurs in mice and humans [38]. Exon 10 splicing depends on the activity of various SR proteins and diverse RNA-interacting and RNA/DNA-binding proteins, among other molecules [39]. SR protein kinase (SRPK) and cdc-like kinase (CLK/Sty) phosphorylate SR proteins and control their functions [40,41,42]. Other kinases, such as cAMP-dependent protein kinase (PKA), serine/threonine-specific protein kinase AKT1, and glycogen synthase kinase-3β (GSK-3β), have the additional capacity to phosphorylate selected SR proteins [43,44,45,46].

The present observations in hTau mice inoculated with AD sarkosyl-insoluble fractions show phosphorylation at specific sites of PKA, AKT1, GSK-3β, and CLK1 in neurons, mainly in the CA1 region. Activation of these kinases is a potential mechanism modelling exon 10 splicing in inoculated hTau mice. However, this scenario does not explain the appearance of 3Rtau in inoculated WT.

Several tau fractions in AD have differential prion-like activities [4]. Following this line of thinking, the diversity of tau strains may contribute to clinical diversity in AD [47,48]. As a working hypothesis, it may be postulated that variable composition of 3Rtau and 4Rtau isoforms occurs in different brain regions and in different neurons in a particular individual. Therefore, basal tau variability, in addition to tau strains, would contribute not only to individual regional and cell vulnerability in the context of different tauopathies, but also to the phenotypic variability in individuals affected by the same tauopathy.

## 4. Materials and Methods

### 4.1. Brain Samples

Brain samples of the hippocampus were obtained from the Institute of Neuropathology Brain Bank, Bellvitge University Hospital, following the guidelines of the Spanish legislation on this matter (Real Decreto Biobancos 1716/2011) and the approval of the local ethics committee of the Bellvitge University Hospital (Hospitalet de Llobregat, Barcelona, Spain). The agonal state was short with no evidence of acidosis or prolonged hypoxia; the pH of the brains was between 6.8 and 7. At the time of autopsy, one hemisphere was fixed in paraformaldehyde for no less than 3 weeks, and selected brain sections were embedded in paraffin; de-waxed paraffin sections, 4 microns thick, were processed with neuropathological and immunohistochemical methods as detailed elsewhere [49]. The other hemisphere was cut into coronal sections 1 cm thick, and selected brain regions were dissected, immediately frozen at −80 °C, put in labeled plastic bags, and stored at −80 °C until use; the rest of the coronal sections were frozen and stored at −80 °C [24]. Paraffin sections were used as controls of positive staining in immunohistochemical techniques.

### 4.2. Extraction of Sarkosyl-Insoluble Fractions and Western Blotting

Frozen samples of the hippocampus from one AD case (male, 68 years old; Braak and Braak NFT stage V-VI; Braak β-amyloid stage C; Thal phase 4; and CERAD 3) and one age-matched control case with no lesions (Braak 0/0, Thal 0; CERAD 0) were processed in parallel. The AD and the control cases did not show concomitant pathology and co-morbidities; in particular, argyrophilic grains, thorn-shaped astrocytes, and coiled bodies were absent. Frozen samples of about 1g were lysed in 10 volumes (*w*/*v*) with cold suspension buffer (10 mM Tris-HCl, pH 7.4, 0.8 M NaCl, 1 mM EGTA) supplemented with 10% sucrose, protease, and phosphatase inhibitors (Roche). The homogenates were first centrifuged at 20,000× *g* for 20 min (Ultracentrifuge Beckman with 70Ti rotor), and the supernatant (S1) was saved. The pellet was re-homogenized in five volumes of homogenization buffer and re-centrifuged at 20,000× *g* for 20 min. The two supernatants (S1 + S2) were then mixed and incubated with 0.1% N-lauroylsarkosynate (sarkosyl) for 1 h at room temperature while being shaken. Samples were then centrifuged at 100,000× *g* for 1 h. Sarkosyl-insoluble pellets (P3) were re-suspended (0.2 mL/g) in 50 mM Tris–HCl (pH 7.4). Protein concentrations were quantified with the bicinchoninic acid assay (BCA) assay (Pierce). Sarkosyl-insoluble fractions were processed for Western blotting. Samples were mixed with loading sample buffer and heated at 95 °C for 5 min. Sixty µg of protein was separated by electrophoresis in SDS-PAGE gels and transferred to nitrocellulose membranes (200 mA per membrane, 90 min). The membranes were blocked for 1 h at room temperature with 5% non-fat milk in TBS containing 0.2% Tween and were then incubated with the phospho-specific anti-tau Ser422 antibody (tau-P-Ser422) (diluted 1:1000; Thermo Fisher Scientific, Barcelona, Spain). After washing with TBS-T, blots were incubated with the appropriate secondary antibody (anti-rabbit IgG conjugated with horseradish peroxidase diluted at 1:2000, Agilent, Barcelona, Spain) for 45 min at room temperature. Immune complexes were revealed by incubating the membranes with chemiluminescence reagent (Amersham) [22,23,26].

### 4.3. Animals

The experiments were carried out on WT mice, homozygous transgenic mice knock-out for murine tau protein (mtWT), and heterozygous mice expressing human isoforms of tau protein (hTau; B6.Cg- (GFP)Klt Tg(*MAPT*)8cPdav/J) in a C57BL/6 background. hTau mice were generated by random insertion of the human microtubule-associated protein tau (*MAPT)* gene into ‘tau knockout’ mice (mtWT) that have a neomycin cassette-induced disruption at exon 1 of the endogenous mouse *Mapt*.33,34; *MAPT* transgene is driven by a tau promoter that causes these mice to over-express human 3Rtau and 4Rtau in the absence of endogenous mouse *Mapt*) [27,28]. Transgenic mice were identified by genotyping genomic DNA isolated from tail clips using the polymerase chain reaction (PCR) conditions indicated by Jackson Laboratory (Bar Harbor, ME, USA). Animals were maintained under standard animal housing conditions in a 12-h dark-light cycle with free access to food and water. All animal procedures were carried out following the guidelines of the European Communities Council Directive 2010/63/EU and with the approval of the local ethical committee (C.E.E.A: Comitè Ètic d’Experimentació Animal; University of Barcelona, Spain; ref. 426/18). Animals were killed by decapitation and their brains were then rapidly removed and processed for study. The left cerebral hemisphere, brainstem, and cerebellum were dissected on ice, immediately frozen, and stored at −80 °C until used for biochemical studies. The right hemisphere, brainstem, and cerebellum were fixed in 4% paraformaldehyde, cut in coronal sections, and embedded in paraffin. De-waxed sections were stained with hematoxylin and eosin, or processed for immunohistochemistry. The immunohistochemical characterization of mice was carried out at 6 and 9 months, and the biochemical study in mice aged 9 months; four animals per group were assessed (total: 24).

### 4.4. (Phospho)Proteomics of Sarkosyl-Insoluble Fractions

Protein extracts were precipitated with the ReadyPrep™ 2-D Cleanup Kit (BioRad, Hercules, CA, USA) and pellets dissolved in 6M Urea and Tris 100 mM pH 7.8. Protein quantitation was performed with the Bradford assay kit (Bio-Rad). A total of 600 µg of protein was used to obtain the phosphorylated fractions from PHFs. For protein digestion, reduction was performed by addition of DTT to a final concentration of 10mM and incubation at RT for 30 min. Subsequent alkylation by 30 mM (final concentration) iodoacetamide was performed for 30 min in the dark at room temperature. An additional reduction step was performed with 30 mM DTT (final concentration), allowing the reaction to stand at room temperature for 30 min. The mixture was diluted to 0.6 M urea using MilliQ-water, and after addition of trypsin (Promega) (enzyme:protein, 1:50 *w*/*w*), the sample was incubated at 37 °C for 16 h. Digestion was quenched by acidification (pH < 6) with acetic acid. After protein enzymatic cleavage, peptide cleaning was performed using Pierce™ Peptide Desalting Spin Columns (ThermoFisher). The enrichment of phosphorylated peptides was performed applying the SIMAC protocol as previously described [50]. Phosphorylated and non-modified peptide fractions were dried in a vacuum and reconstituted into a final concentration of 0.5 µg/µL of 2% ACN, 0.5% FA, 97.5% MilliQ-water prior to mass spectrometric analysis. Samples were first loaded for concentration into an Acclaim™ PepMap™ 100 C18 trap column (0.1 mm × 20 mm, particle size 5 µm; ThermoFisher). Mobile phases were 100% water 0.1% formic acid (FA) (buffer A) and 100% Acetonitrile 0.1% FA (buffer B). Column gradient was developed from 2% B to 40% B in 120 min. Column was equilibrated in 95% B for 10 min and 2% B for 10 min. During the entire process, precolumn was in line with column and flow maintained all along the gradient at 300 nl/min. Eluting peptides were analyzed using a 5600 Triple-TOF mass spectrometer (Sciex). Information data acquisition was made on a survey scan performed in a mass range from 350 m/z up to 1250 m/z in a scan time of 25 ms. The top 15 peaks were selected for fragmentation. Minimum accumulation time for MS/MS was set to 200 ms giving a total cycle time of 3.3 s. Product ions were scanned in a mass range from 100 m/z up to 1500 m/z and excluded for further fragmentation during 15 s. The raw MS/MS spectra searches were processed using the MaxQuant software (v 1.6.7.0, Max Planck Institute of Biochemistry, Planegg, Germany) and searched against the Uniprot proteome reference for Homo sapiens (Proteome ID: UP000005640_9606). The parameters used were as follows: initial maximum precursor (15 ppm) fragment mass deviations (20 ppm); fixed modification (Carbamidomethyl (C)); variable modification (Oxidation (M); Acetyl (Protein N-terminal; Deamidation (N); Gln > puroGlu; Phospho (STY)); enzyme (trypsin) with a maximum of one missed cleavage; minimum peptide length (seven amino acids); and false discovery rate (FDR) for PSM and protein identification (1%). Frequently observed laboratory contaminants were removed.

### 4.5. Inoculation of Sarkosyl-Insoluble from Alzheimer Disease and Control Homogenates in the Hippocampus of WT, hTau, and mtWT Mice

Mice were inoculated with the same AD sarkosyl-insoluble fraction to avoid differences related to the composition of the inoculum. One series of mice was inoculated at the age of 6 months and killed at the age of 9 months (*n* = 6 per group); the other series was inoculated at the age of 3 months and killed at the age of 9 months (*n* = 6 per group) (Total: 36). Other groups of mice were inoculated with sarkosyl-insoluble fractions from control homogenates at the age of 6 months and killed at the age of 9 months (*n* = 4 per group). No group showed differences in survival times due to the inoculum. One WT mouse was inoculated with vehicle (50 mM Tris-HCl, pH 7.4) at the age of 3 months and killed at the age of 9 months. Mice were deeply anesthetized with an intra-peritoneal injection of ketamine/xylazine/buprenorphine cocktail and placed in a stereotaxic frame after assuring lack of reflexes. For the intra-hippocampal inoculation, we used a Hamilton syringe (volume 2.0 µL); the coordinates were −1.9 mm interaural; −1.4 mm relative to Bregma, and −1.8 mm DV from the dural surface [29]. A volume of 1.5 µL was injected at a rate of 0.05 µL/min. The syringe was withdrawn slowly over a period of 10 min to avoid leakage of the inoculum. Following surgery, the animals were kept in a warm blanket and monitored until they recovered from the anesthesia. Carprofen analgesia was administered immediately after surgery and once a day during the next two consecutive days. Animals were housed individually with full access to food and water. Mice were killed by decapitation, and the brains were rapidly fixed with 4% paraformaldehyde in phosphate buffer, cut in coronal sections, and embedded in paraffin.

### 4.6. Immunohistochemistry

De-waxed sections, 4 microns thick, were processed for immunohistochemistry. The sections were boiled in citrate buffer pH = 6 (20 min) to retrieve tau antigenicity. Endogenous peroxidases were blocked by incubation in 10% methanol-1% H_2_O_2_ solution (15 min) followed by 3% normal horse serum solution. The sections were incubated at 4 °C overnight with one of the primary antibodies listed in Table 2. Following incubation with the primary antibody, the sections were incubated with EnVision + system peroxidase (DakoAgilent, Barcelona, Spain, DK) for 30 min at room temperature. The peroxidase reaction was visualized with diaminobenzidine and H_2_O_2_. Control of the immunostaining included omission of the primary antibody; no signal was obtained following incubation with only the secondary antibody. Lesions were assessed in consecutive sections within coordinates −2.3 to −1.6 interaural, and −1.4 to −2.1 from Bregma.

### 4.7. Quantification of Abnormal Deposits Revealed by Immunohistochemistry

Since differences with the majority of markers were manifested in binary terms, positive or negative, no densitometric assays were conducted in the different groups of non-inoculated and inoculated mice. Deposits in neurons and related axonal fibers and threads in the hippocampal complex were expressed semi-quantitatively with the following signs: +++: abundant; ++: moderate; +: scarce; +/−: not present in every case. However, densitometric studies were carried out in sections stained with anti-3Rtau and anti-4Rtau antibodies to identify variations in the levels of the immunostaining.

Photomicrographs of sections of the dentate gyrus (DG) and CA1 stained with anti-3Rau and anti-4Rtau antibodies were obtained at a magnification of ×200, covering an area of 0.126 mm^2^, using a DP25 camera adapted to an Olympus BX50 light microscope. The pictures, two areas per region per case in every case, were analyzed using Photoshop software. Non-inoculated WT, mtWT, and hTau mice, and AD-inoculated hTau mice, surviving six months after injection, were assessed. The density of tau staining was calculated as the intensity of the diaminobenzidine (DAB) precipitate pigment normalized for the total area and expressed as a percentage of arbitrary units per area. Kolmogorov–Smirnov test was used to test the normality of the distribution. Results were analyzed with one-way ANOVA and post hoc Tukey. Differences between groups were considered statistically significant at *** *p* < 0.001 when comparing WT, and ### *p* < 0.001 when comparing to mtWT.

### 4.8. Double-Labeling Immunofluorescence and Confocal Microscopy

De-waxed sections, 4 microns thick, were stained with a saturated solution of Sudan black B (Merck) for 15 min to block autofluorescence of lipofuscin granules present in cell bodies, and then rinsed in 70% ethanol and washed in distilled water. The sections were boiled in citrate buffer to enhance antigenicity and blocked for 30 min at room temperature with 10% fetal bovine serum diluted in PBS. Then, the sections were incubated at 4 °C overnight with combinations of primary antibodies against different proteins. The characteristics of the antibodies, the dilutions, and the suppliers are listed in Table 2. After washing, the sections were incubated with Alexa488 or Alexa546 (1:400, Molecular Probes) fluorescence secondary antibodies against the corresponding host species. Nuclei were stained with DRAQ5™ (1:2000, Biostatus). After washing, the sections were mounted in Immuno-Fluore mounting medium (ICN Biomedicals), sealed, and dried overnight. Sections were examined with a Leica TCS-SL confocal microscope.

### 4.9. Gel Electrophoresis and Western Blotting

Frozen samples of the whole mouse brain were homogenized in RIPA lysis buffer composed of 50 mM Tris/HCl buffer, pH 7.4 containing 2 mM EDTA, 0.2% Nonidet P-40, 1 mM PMSF, protease and phosphatase inhibitor cocktail (Roche Molecular Systems, Basel, Switzerland). The homogenates were centrifuged for 20 min at 12,000 rpm. Protein concentration was determined with the BCA method (Thermo Scientific). Equal amounts of protein (12 μg) for each sample were loaded and separated by electrophoresis on 10% sodium dodecyl sulfate polyacrylamide gel electrophoresis (SDS-PAGE Invitrogen, Thermo Fisher) gels and transferred onto nitrocellulose membranes (Amersham, Buckinghamshire, UK). Non-specific bindings were blocked by incubation in 3% albumin in PBS containing 0.2% Tween for 1 h at room temperature. After washing, membranes were incubated overnight at 4 °C with antibodies against different forms of tau protein (Table 2). Protein loading was monitored using an antibody against β-actin (42 kDa, 1:30,000, Sigma). Membranes were incubated for 1 h with appropriate HRP-conjugated secondary antibodies (1:3000, Dako); the immunoreaction was revealed with a chemiluminescence reagent (ECL, Amersham). Densitometric quantification was carried out with the ImageLab v4.5.2 software (BioRad), using β-actin for normalization. The brains of four P301S, four hTau, three WT, and three mtWT mice were processed for gel electrophoresis and Western blotting.

### 4.10. Statistical Analysis

Results were analyzed statistically with SPSS 19.0 (SPSS Inc, Chicago, IL, USA) software and GraphPad PRISM software version 9.3.1. (GraphPad Software Inc, San Diego, CA, USA) Data were presented as mean ± standard error of the mean (SEM). Data were compared with one-way analysis of variance (ANOVA) followed by post hoc Tukey’s multiple comparison test or Student’s *t*-test when necessary. Significance level was set at * *p* < 0.05, ** *p* < 0.01, *** *p* < 0.001 vs. Wt; # *p* < 0.05, ### *p* < 0.001 vs. P301S; and $ *p* < 0.05; $$ p < 0.01 and $$$ *p* < 0.001 vs. mtWt.

## Figures and Tables

**Figure 1 ijms-23-00718-f001:**
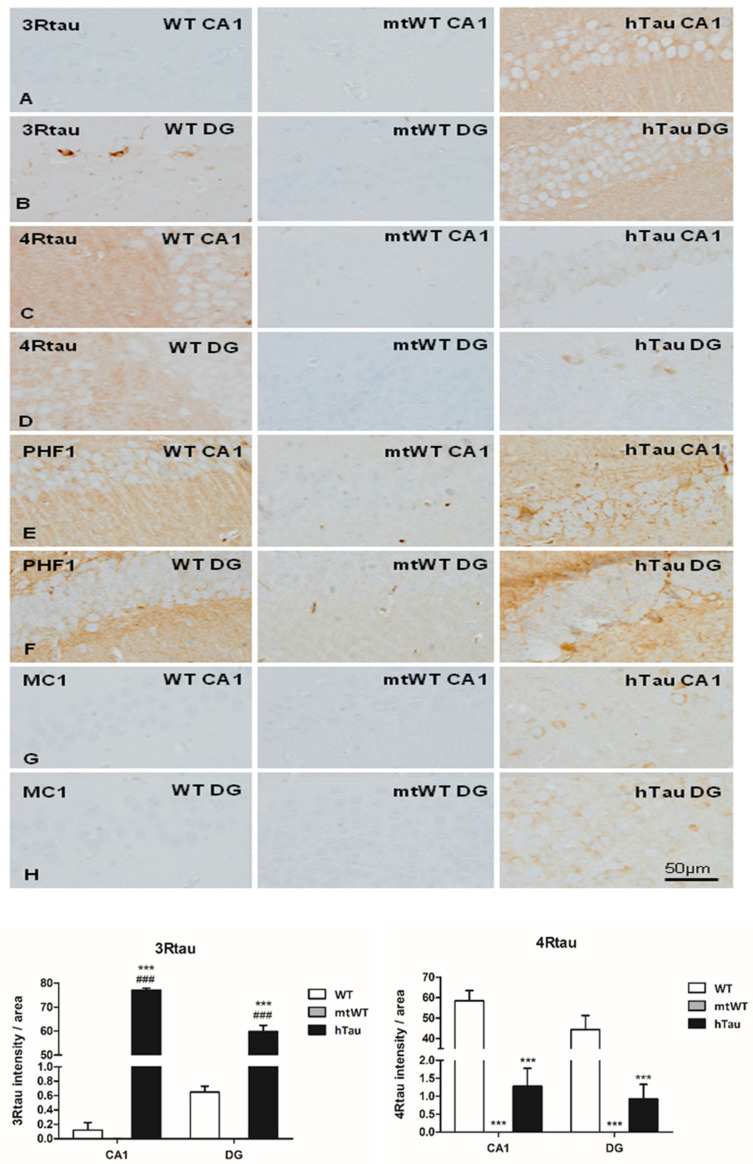
Representative images of 3Rtau (**A**,**B**), 4Rtau (**C**,**D**), PHF1 (**E**,**F**), and MC1 (**G**,**H**) expression in the CA1 region of the hippocampus and dentate gyrus (**GD**) in WT, mtWT, and hTau at the age of 9 months. In hTau mice, 3Rtau predominates, whereas only 4Rtau is expressed in WT mice. Weak PHF1 immunoreactivity occurs in the neuropil in WT, but PHF1-immunoreactive neurons are only seen in hTau mice. MC1 antibodies decorate weak globular cytoplasmic deposits in a subpopulation of neurons in hTau mice. MC1 is negative in WT. As expected, immunohistochemistry with antibodies against 3Rtau, 4Rtau, PHF1, and MC1 is negative in mtWT transgenic mice. Paraffin sections with slight haematoxylin counterstaining, bar = 50 µm. Graphs show quantitative densitometry for anti-3Rtau and anti-4Rtau immunostaining expressed as a percentage of arbitrary units per area in CA1 and dentate gyrus (**DG**) regions in WT, mtWT, and hTau mice. Results were analyzed with one-way ANOVA and post hoc Tukey. Differences between groups were considered statistically significant at *** *p* < 0.001 when comparing hTau with WT, and ### *p* < 0.001 when comparing htau with mtWT. Note that 4Rtau is predominant in WT mice, excepting the DG in which 3Rtau positive neurons are also localized in the inner layer. In contrast 3Rtau predominates in CA1 and DG in hTau transgenic mice.

**Figure 2 ijms-23-00718-f002:**
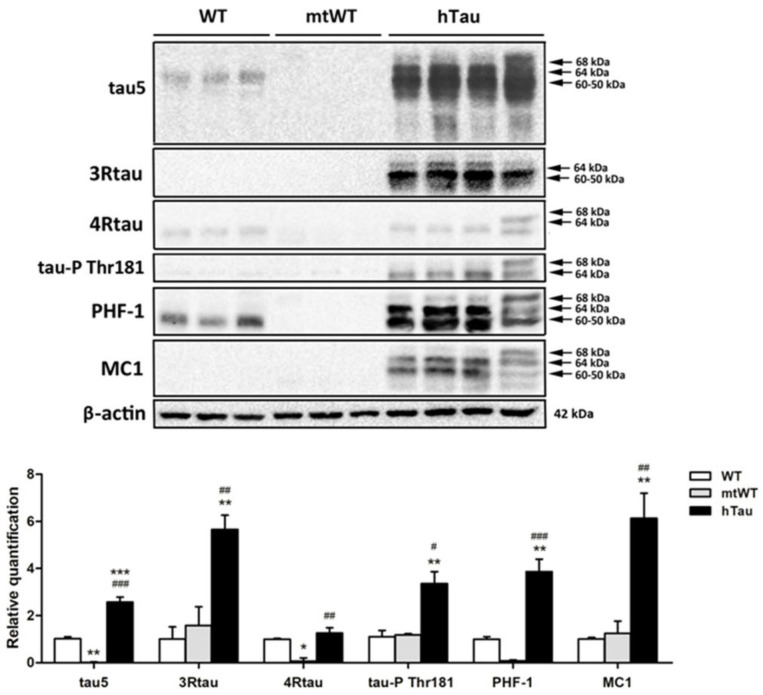
Gel electrophoresis and Western blotting of total brain homogenates from WT, mtWT, and hTau transgenic mice at the age of 9 months processed with Tau 5, 4Rtau, 3Rtau, phosphorylated tau at Thr181 (tau-Pthr181), PHF1, and MC1 antibodies, and processed in parallel. β-actin was used as a control of protein loading. Since the profile of β-actin was similar in the different Western blots, only one representative example is included in the figure. WT mice show the presence of total tau (tau 5), mainly composed of 4Rtau. A PHF1-immunoreactive band is also observed in WT mice; the presence of this band correlates with PHF1 immunoreactivity in the neuropil in these mice. Western blots using tau-P, Thr181, and MC1 antibodies are negative. hTau transgenic mice express a large amount total tau manifested in sbands of 68 kDa, 64 kDa, and 60–50 kDa accompanied by an upper band and several lower bands of smears and truncated tau. It can be seen that 3Rtau is more abundant than 4Rtau; bands of 64 kDa and 60–50 kDa are detected with the 3Rtau antibody, and bands of 68 kDa and 64 kDa with the 4R antibody. PHF1 and MC1 antibodies recognize several bands of about 68 kDa, 64 kDa, and 60–50 kDa. mtWT mice lack tau of any species. The molecular weights labeling tau bands are approximate. Since the samples were run in parallel the molecular weights in the right are also valid for the lanes of WT and mtWT homogenates. The graph shows the relative quantification with Western blotting of tau protein levels in total brain homogenate of WT, mtWT, and hTau mice aged 9 months. One-way analysis of variance (ANOVA) followed by Tukey test. Data are expressed as the mean values ± SEM. The significance level was set at * *p* < 0.05 ** *p* < 0.01, *** *p* < 0.001 vs. WT and # *p* < 0.05, ## *p* < 0.01, ### *p* < 0.001 vs. mtWT.

**Figure 3 ijms-23-00718-f003:**
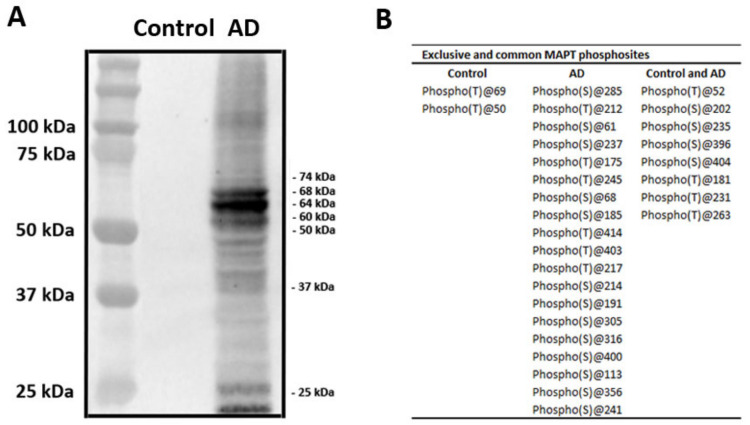
Characteristics of sarkosyl-insoluble fractions. (**A**) Sarkosyl-insoluble fractions of AD brain homogenates blotted with anti-tauSer422 were characterized by three bands of 68 kDa, 64 kDa, and 60 kDa, a weak upper band of 73 kDa, and several lower bands of fragmented tau between 50 and 25 kDa. Lower bands stained with anti-tauSer422 show truncated tau at the C-terminal. Sarkosyl-insoluble fractions of the control case blotted with the same antibody are negative. The first lane on the left shows the molecular weight of the crude markers. (**B**) Table showing the phosphorylation sites in tau from sarkosyl-insoluble fractions of AD and control homogenates revealed by (phospho)proteomics. The phosphorylation site is labeled with T for threonine and S for serine. The table shows two phosphorylation sites in tau found only in the control; nineteen phosphorylation sites found only in AD; and eight phosphorylation sites which were common to control and AD. (phospho)proteomics was used to learn about the molecular composition of sarkosyl-insoluble fractions in AD and control cases. Twenty-seven phosphorylated tau sites were identified in AD, and ten phosphorylated tau sites in the control; eight phosphosites were shared by AD and control homogenates (**B**).

**Figure 4 ijms-23-00718-f004:**
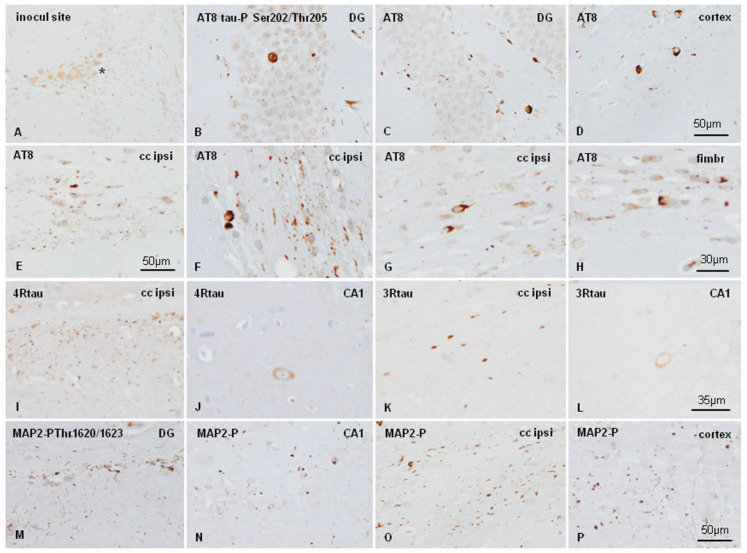
WT mice unilaterally inoculated into the hippocampus with AD sarkosyl-insoluble fractions at age six months and killed at the age of nine months showed a few macrophages filled with debris at the site of inoculation (**A**), and tau deposits in the **CA1** region of the hippocampus (**CA1**), dentate gyrus (**DG**), cerebral cortex (cortex), ipsilateral corpus callosum cc (ipsi), and fimbria (fimbr). Strong AT8 immunoreactivity is seen in neurons of the DG (**B**,**C**), and cortex near the site of inoculation (**D**), and in threads and glial cells in the corpus callosum (**E**). At higher magnification AT8-immunoreactive inclusions in glial cells in the corpus callosum and fimbria (**F**–**H**) are reminiscent of coiled bodies. Deposits of 4Rtau and 3Rtau are found in threads in the corpus callosum (**I** and **K**, respectively), and in neurons of the CA1 region (**J** and **L**, respectively). In addition to tau deposits, abnormal inclusions also recognized with anti-MAP2-P Thr1620/1623 antibodies (**M**–**P**). Paraffin sections slightly counterstained with hematoxylin, (**A**–**E**),(**M**–**P**), bar = 50 µm; (**F**–**H**), bar =30 µm; (**J**–**L**), bar = 35 µm.

**Figure 5 ijms-23-00718-f005:**
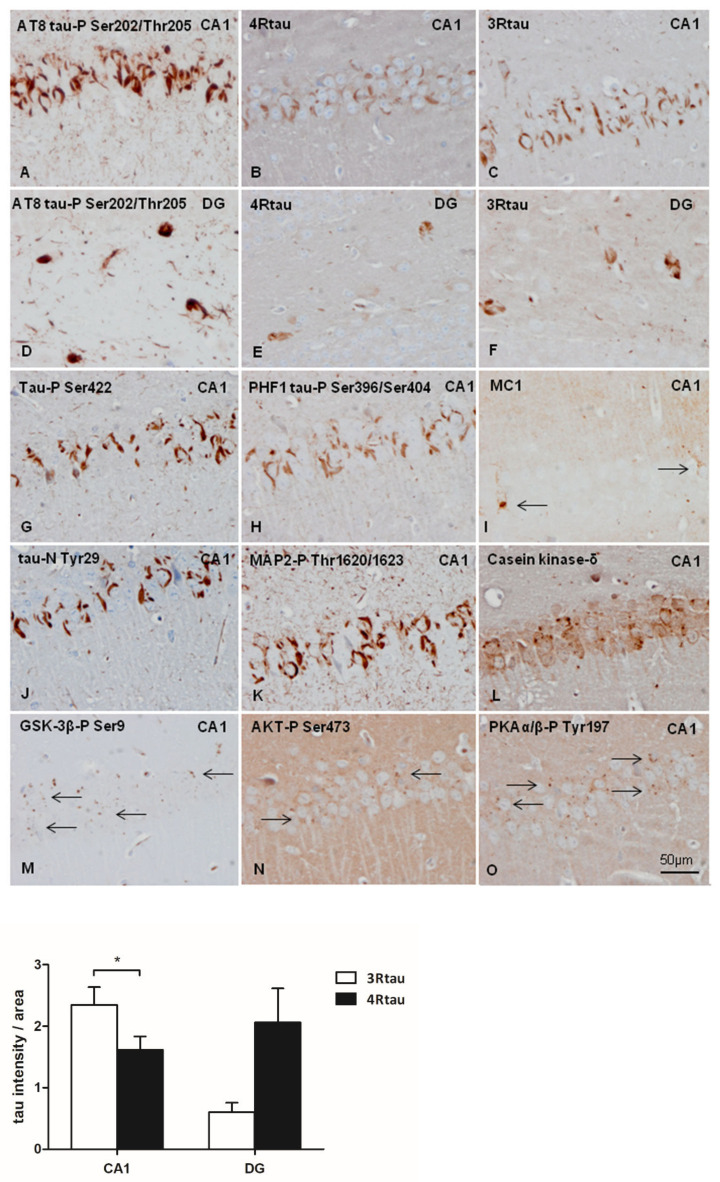
hTau transgenic mice unilaterally inoculated in the hippocampus with sarkosyl-insoluble fractions from AD homogenates at the age of six months and killed at the age of nine months show abundant abnormal deposits in the CA1 region of the hippocampus (**CA1**) and dentate gyrus (**DG**). AT8-immunoreactive deposits are seen in CA1 region (**A**), and dentate gyrus and hilus (**D**). Tau deposits are stained with anti-4Rtau antibodies (**B**,**E**), and more strongly with anti-3Rtau antibodies (**C**,**F**). Phospho-tau deposits in hTau transgenic mice are dense and flame-shaped, reminiscent of neurofibrillary tangles. Immunoreactivity to tau-P Ser422, PHF1, MAP2-P, and tau-N Tyr29 is strongly positive (**G**, **H**, **K** and **J**, respectively). MC1 immunoreactivity is restricted to a few neurons (**I**); in this field, one neuron has small granular deposits (right arrow) and the other a dense round inclusion (left arrow). CK1-δ (**L**), GSK-3β-P Ser9 (**M**), AKT-P Ser473 (**N**), and PKAα/β-P Tyr197 (**O**) granular immunoreactivity (arrows) appears in affected neurons in inoculated hTau mice. Sections are consecutive and obtained from the same inoculated hTau mouse. Paraffin sections slightly counterstained with haematoxylin, bar = 50 µm. The graph shows quantitative densitometry for anti-3Rtau and anti-4Rtau staining expressed as a percentage of arbitrary units per area in CA1 and dentate gyrus (DG) regions in inoculated hTau mice surviving six months after injection. Results were analyzed with Student’s *t*-test. Differences between anti-3Rtau and anti-4Rtau staining within the same region in consecutive slices were considered statistically significant at * *p* < 0.05.

**Figure 6 ijms-23-00718-f006:**
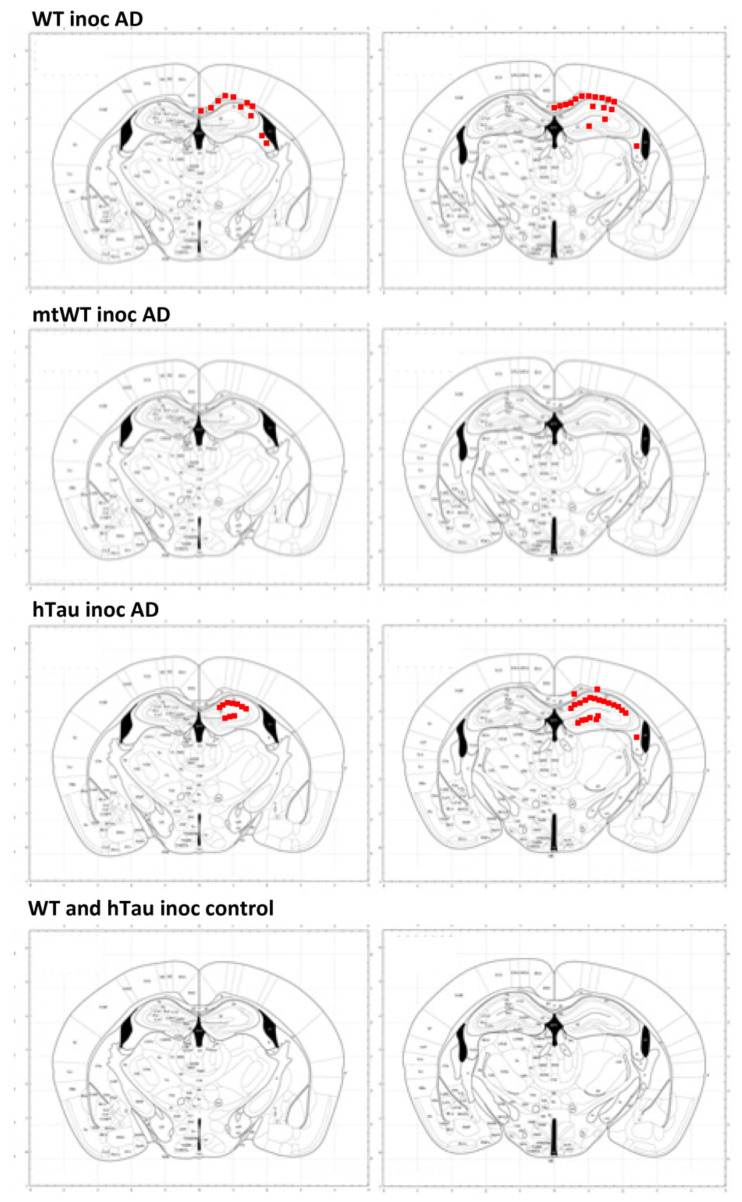
The distribution of abnormal tau deposits in WT, mtWT, and hTau unilaterally inoculated in the hippocampus with the same inocula of sarkosyl-insoluble fraction of AD and WT, and hTau inoculated with sarkosyl-insoluble fractions from one control brain. Mice were inoculated at the age of six months and killed at the age of nine months. Red points are tau deposits representative of deposits depicted in Figure 4 and Figure 5. Tau deposits are not detected in mtWT inoculated with AD sarkosyl-insoluble fractions. In contrast, Tau deposits are found in WT and hTau inoculated with AD sarkosyl-insoluble fractions. Tau deposits in CA1 region and dentate gyrus are more abundant in inoculated hTau mice when compared with WT mice. Tau deposits in the stratum radiatum and stratum oriens are also more abundant in hTau than in WT mice. In contrast, tau deposits in the corpus callosum and fimbria (involving glial cells) are more copious in inoculated WT mice. No abnormal tau deposits are detected in WT and hTau mice inoculated with sarkosyl-insoluble fractions from a control case. The maps were obtained from the Atlas of Paxinos and Franklin [29] and represent two different coronal levels to facilitate the extent of tau deposits considering the three axes; sections of the left column are more anterior than the sections of the right column.

**Figure 7 ijms-23-00718-f007:**
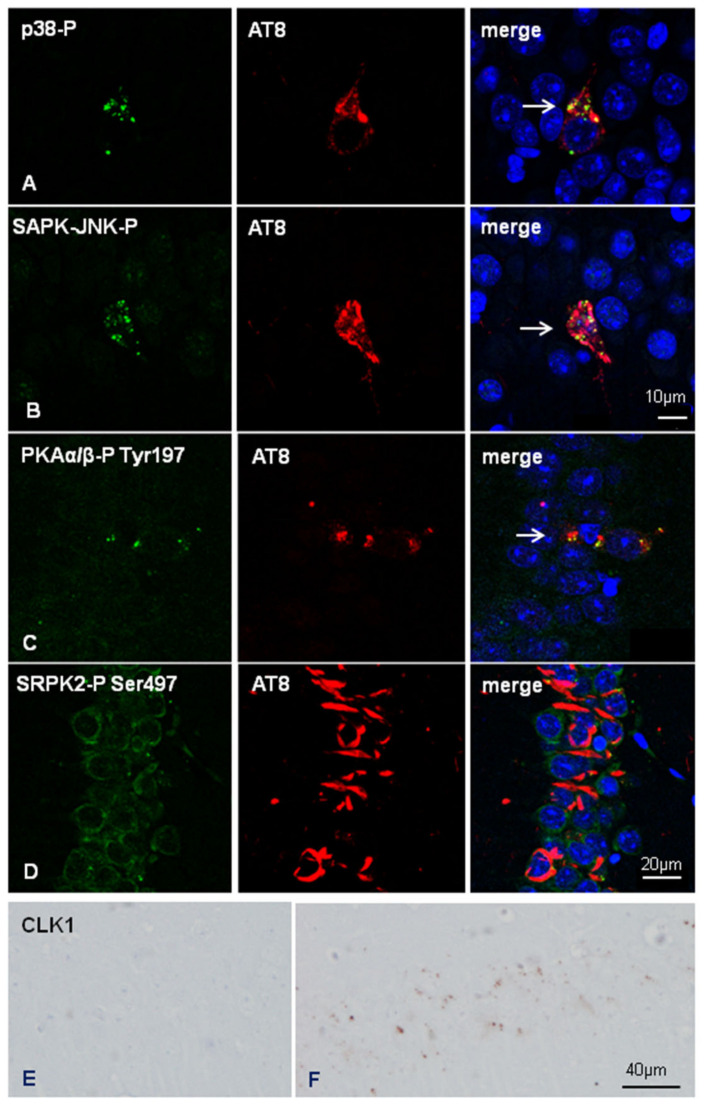
Double-labeling immunofluorescence and confocal microscopy to p38-P Thr180/Tyr182 or SAPK/JNK-PThr183/Thr185 (green), and AT8 (red) in inoculated hTau at the age of six months and killed at the age of nine months shows co-localization of the active kinases with phospho-tau deposition (white arrows) (**A** and **B**, respectively for p38-P and SAPK-JNK-P) in dentate gyrus neurons. Double-labelling immunofluorescence and confocal microscopy to PKAα/β-P Tyr197 and AT8 in inoculated hTau mice shows cytoplasmic granular PKAα/β-P Tyr197 granules in CA1 neurons with abnormal phospho-tau deposits in neurons (**C**, arrow). Cytoplasmic SRPK2-Ser497 immunoreactivity is found in neurons with phospho-tau deposits in inoculated hTau mice (**D**). Paraffin sections, nuclei are stained with DRAQ5™ (blue), A, B, bar = 10 µm, C and D, bar = 20 µm. CLK1 immunoreactivity in the CA1 region of non-inoculated (**E**) and inoculated hTau with AD sarkosyl-insoluble fractions (**F**); small cytoplasmic granules are seen in inoculated hTau. Paraffin sections, slightly counterstained with haematoxylin, bar = 40 µm.

**Table 1 ijms-23-00718-t001:** Characteristics of tau deposits in neurons of the hippocampus and dentate gyrus in non-inoculated WT, hTau, and mtWT mice; WT, and hTau mice unilaterally inoculated into the hippocampus with the same AD inocula (inoc AD WT, and inoc AD tau, respectively); and inoculated with sarkosyl-insoluble fractions from controls (inoc C hTau). nft: neurofibrillary tangles, ptg: pre-tangles; gr: granular deposits; gl: globular deposits. EC: entorhinal cortex. Age of non-inoculated and inoculated mice at the time of examination: 9 months. The results are expressed semi-quantitatively considering the whole tissue per case in every case; the following signs indicate: +++: abundant; ++: moderate; +: scarce; +/−: not present in every case.

Antibody	WT	hTau	mtTau	Inoc AD WT	inoc AD Tau	Inoc C hTau
3Rtau	-	diffuse	-	+	++ (nft)	diffuse
4Rtau	diffuse	diffuse	-	+	+++ (nft)	diffuse
AKT-P Ser473	-	-	-	-	++ (gr)	-
CLK1	-	-	-	-	++ (gr)	-
CK1-δ	-	-	-	-	++ (gr)	-
eIF2α-P Ser51	-	-	-	-	-	-
GSK-3β-P Ser9	-	-	-	-	++ (gr)	-
IRE-P Ser274	-	-	-	-	-	-
LAMP1	-	-	-	-	-	-
LC3	-	-	-	-	-	-
MAP2-P Thr1620/1623	-	-	-	+ (ptg)	+++ (nft)	-
NFL-P Ser473	-	-	-	-	-	-
p38-P Thr180/Tyr182	-	-	-	+ (gr)	++ (gr)	-
PHF1, tau-P Ser396/Ser404	-	+/− (gl)	-	-	++ (nft)	+/− (gl)
PKAα/β-P Tyr197	-	-	-	-	++ (gr)	-
SAPK/JNK-P-Thr183/Thr185	-	-	-	+ (gr)	++ (gr)	-
SRC-P Tyr416	-	-	-	-	-	-
SRPK1	-	-	-	-	-	-
SRPK2-P Ser497	-	-	-	-	-	-
tau AT8-P Ser202/Thr205	nuclei	-	-	+ (ptg)	+++ (nft)	-
tau C-3	-	-	-	-	-	-
tau MC1	-	+/− (gl)	-	+ (ptg)	+++ (nft)	+/− (gl)
tau-N Tyr29	-	-	-	-	++ (nft)	-
tau-P Ser422	-	-	-	+ (ptg)	+++ nft)	-
Ubiquitin	-	-	-	-	-	-

**Table 2 ijms-23-00718-t002:** List of antibodies used for immunohistochemistry and Western blotting; 3Rtau: tau with three repeats; 4Rtau: tau with four repeats; AKT1: serine/threonine-protein kinase AKT1; β-actin; CK1-δ: casein kinase δ; eIF2α: eukaryotic initiation factor 2α; GSK β: glycogen synthase kinase 3β; IRE1: inositol-requiring enzyme 1α; LAMP1: lysosomal-associated membrane protein 3; LC3: microtubule-associated protein 1A/1B-light chain 3; MAP2: microtubule-associated protein 2; NFL: neurofilament light molecular weight; p38: p38-kinase; PHF1: anti-paired helical filament 1; PKAα/β: cAMP-dependent protein kinase A; SAPK/JNK: stress-activated protein kinase/Jun amino-terminal kinase; SRC: tyrosine-protein kinase SRC; Tau 100; Tau 5: monoclonal recognizing total tau protein; Tau AT8; Tau C3: tau truncated at aspartic acid 421; Tau MC1: conformation tau; Tau-N: nitrated tau; Tau-P phosphorylated tau. The phosphorylation or nitration of the protein at the specific site is indicated in every case. WB dil: Western blotting dilution; IHQ dil: immunohistochemistry dilution.

Antibody	Supplier	Reference	Host	WB Dil	IHQ Dil
3Rtau	Upstate (Syracuse, NY, USA)	05-803	Ms	1/1000	1/800
4Rtau	Millipore (Barcelona, Spain)	05-804	Ms	1/1000	1/50
AKT1-P Ser473	Abcam (Cambridge, UK)	Ab18206	Rb	-	1:100
β-actin	Sigma (Barcelona, Spain)	A5316	Ms	1/30,000	-
CLK1 (75-100 aa)	LS Bio (Derio, Spain)	LS-C382760	Ms	-	1/100
CK1-δ	Abcam (Cambridge, UK)	ab85320	Ms	-	1/500
eIF2α-P Ser51	Thermo Scientific (Barcelona, Spain)	MA5-15133	Rb	-	1/100
GFAP	Dako (Barcelona, Spain)	Z0334	Rb	-	1/400
GSK-3β-P Ser9	Cell signaling (Danvers**,** MA, USA)	9336	Rb	-	1/100
Iba1	Wako (Richmond, VA, USA)	019-19741	Rb	-	1/1000
IRE-P Ser274	Abcam (Cambridge, UK)	ab48187	Rb	-	1/100
LAMP1	Santa Cruz (CA, USA)	Sc5570	Rb	-	1/10
LC3	Cell Signaling (Danvers, MA, USA)	2775	Rb	-	1/100
MAP2-P Thr1620/1623	Cell Signaling (Danvers, MA, USA)	4544	Rb	-	1/1000
NFL-P Ser473	Millipore (Burlington, MA, USA)	MABN2431	Ms	-	1/100
p38-P Thr180/Tyr182	Cell Signaling (Danvers, MA, USA)	9211	Rb	-	1/100
PHF1, tau-P Ser396/Ser404	Dr. Peter Davies	-	Ms	-	1/500
PKAα/β-P Tyr197	Invitrogen (Carlsbad, CA, USA)	44988	Rb		1/100
SAPK/JNK-P-Thr183/Thr185	Cell Signaling (Danvers, MA, USA)	9251	Rb	-	1/25
SRC-P Tyr416	Millipore (Barcelona, Spain)	05-677	Ms	-	1/100
SRPK1	Abcam (Cambridge, UK)	ab90527	Rb	-	1/100
SRPK2-P Ser497	Affbiotech (Bionova, Spain)	3632	Rb	-	1/100
tau 5	Thermo Scientific (Barcelona, Spain)	MA5-12808	Ms	1/500	-
tau AT8-P Ser202/Thr205	Innogenetics (Barcelona, Spain)	90206	Ms	-	1/50
tau C-3	Millipore (Burlington, MA, USA)	36-017	Ms	-	1/100
tau MC1	Dr. Peter Davies	-	Ms	-	1/20
tau-N Tyr29	Millipore (Burlington, MA, USA)	MAB2244	Ms	-	1/200
tau-P Ser422	Thermo Scientific (Barcelona, Spain)	44764	Rb	-	1/50
tau-P Thr181	Thermo Scientific (Barcelona, Spain)	PA1-14413	Rb	1/1000	-
Ubiquitin	Dako (Barcelona, Spain)	Z0458	Ms	-	1/250

## Data Availability

All the supporting data are in the manuscript.

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
