# Peer review of "Host Tau Genotype Specifically Designs and Regulates Tau Seeding and Spreading and Host Tau Transformation Following Intrahippocampal Injection of Identical Tau AD Inoculum"

_ijms, 2022, doi:10.3390/ijms23020718_

Round 1

Reviewer 1 Report

The authors demonstrate that there must be preformed or preforms of tau proteins (vulnerable cells) in the host brain to trigger further tau formation. In this respect I recommend that the authors discuss more extensive the formation of tau proteins out of the seeds in vulnerable cells (unfolded protein response, "stamping" of the conformational states etc.). 

Author Response

The authors demonstrate that there must be preformed or preforms of tau proteins (vulnerable cells) in the host brain to trigger further tau formation. In this respect I recommend that the authors discuss more extensive the formation of tau proteins out of the seeds in vulnerable cells (unfolded protein response, "stamping" of the conformational states etc.).  he authors of this article reported the importance of host tau in tau seeding. The results have the potential to make significant contribution to the literature of tau seeding. However the lack of quantitative interpretation of the immunoreactivity microscope images, the insufficient description of methodology supporting the presented figures and the abundance of typo suggest much room to improve. The followings are my observations:

(unfolded protein response, "stamping" of the conformational states etc.). 

The following text has been added in the discussion

Previous studies have shown the activation of the unfolded protein response in pre-tangle neurons, and the induction of granulovacuolar degeneration bodies (GVBs) following intracerebral inoculation of tau fibrils [34, 35]. Seeding of tau pathology in vivo using synthetic K18 tau P301L PHFs seeds into the hippocampus of tau P301L Tg and induces granulovacuolacuolar degeneration. Similarly, brain lysates of tau P301S Tg mice with filamentous tau injected into the hippocampus and overlaying cerebral cortex of WT tau ALZ17 mice, and injection with human post-mortem brain lysates from donors with confirmed neuropathology of AD and primary tauopathy induces GVBs immunoreactive for CK1δ and pPERK [35]. hTau mice, but not WT mice, inoculated with sarkosyl-insoluble fractions of AD homogenates show cytoplasmic granules in neurons with cytoplasmic tau deposits; these granules are immunoreactive with anti CK1-δ, CLK1, GSK-3β-PSer9, p38-PThr 180/Tyr182, PKA α/β Tyr197, and SAPK/JNK-P Thr183/Thr185 antibodies. However, abnormal deposits in neurons and glial cells are not stained with anti-eIF2α-P Ser51, IRE-P Ser274, LC3, and LAMP1 antibodies in our models, suggesting that they do not present many principal components of granulovacuolar degeneration (GVD) [36, 37].

Reviewer 2 Report

The authors of this article reported the importance of host tau in tau seeding. The results have the potential to make significant contribution to the literature of tau seeding. However the lack of quantitative interpretation of the immunoreactivity microscope images, the insufficient description of methodology supporting the presented figures and the abundance of typo suggest much room to improve. The followings are my observations:

Line 24:

  • The authors pointed out that the literature is lack of attention to the importance of "the molecular changes associated with the transformation of host tau into abnormal tau." However, the article does not explicitly reported "molecular changes" of host tau. The author may want to contrast their work with the literature in a more accurate way.

Figure 1:

  • The authors may want to introduce some more quantitative ways (such as count of dark pixels) to support the arguments of "only 4Rtau is expressed in WT mice (Line 323)" and "4Rtau is negative in tWT transgenic mice (Line 327)". Reading from the figure in the PDF file, I can barely tell the difference between 4Rtau WT CA1 v.s. mtWT CA1 (row C, 1st and 2nd column), and find it hard to conclude that 4Rtau is expressed in WT mice while not in mtWT.

Figure 5:

  • Similar to Figure 1. The author claimed on Line 451 that "Deposits contained 4Rtau (Figure 5B, E), but 3Rtau was more abundant (Figure 5C and F)." However, 5C are visually similar to 5B, and 5F similar to 5E. It is therefore difficult to come to the same conclusion just by looking at the figures. Some quantitative measures will be helpful.

Figure 2:

  • I had a hard time relating the bar plot with the gel image. In the bar plot, the 'WT' white bars across different x labels look almost of identical height. Did the authors scale the values of 'WT' to be 1? If so the authors may want to consider add bounding boxes between different x labels, or separate them into subplots, to prevent readers from comparing bars in one x label with bars in another.

  • Is the last lane of figure 2 the MW marker? Is the last lane label "hTau" correct? For 4Rtau, why is there two bands in the last lane but three arrows of MW?

Figure 3A: The authors may want to add the MW marker lane.

Figure 3B:

  • The authors may want to use a table instead of a Venn diagram.
  • The font size of the text in 3B is too small to read.
  • The two circles take up a lot of empty space while its diameter or area seems not relevant to the number of phosphorylation sites. The blue circle has 19+8=27 sites, the yellow circle has 8+2=10 sites, but the two have the same size.

Figure 4:

  • The authors may want to explicitly label the length of the scale bar instead of appending them at the figure caption. Doing so can greatly help readers to identify relevant objects in the figures.

Line 499 and Figure 6:

  • It is unclear how the authors draw the brain diagrams. Are they reploted from the literature? If so citation is recommended.
  • It is unclear how the red square marks in Figure 6 were decided. Is Figure 6 a replot of both Figure 5 and Figure 4?
  • There seem to be some labels of the left brain in every figure of the Figure 6 but they are unreadable.
  • There are missing labels between the two columns. Are they independent repeats?
  • The 2nd row "mtWT inoc AD" and the 4th row "WT and hTau inoc control" look identical. If there is no any other information to read from the above 2nd and the 4th row except that "there is no deposit", the authors may want to replace the 2nd and the 4th rows with text "Deposit not detected".

Line 521, Figure 7:

  • The authors claimed "In contrast, cytoplasmic SRPK2-Ser497 immunoreactivity is found equally in neurons with and without phospho-tau deposits in hTau mice (D). " Is Figure 7D WITH or WITHOUT phospho-tau deposits?

Typo:

  • Line 327: 'tWT' -> 'mtWT'.
  • Table 2 header: 'inoc AD hau' -> 'inoc AD tau'
  • Line 480: Are the question marks typos?

Author Response

Reviewer 2:

Line 24:

The authors pointed out that the literature is lack of attention to the importance of "the molecular changes associated with the transformation of host tau into abnormal tau." However, the article does not explicitly reported "molecular changes" of host tau. The author may want to contrast their work with the literature in a more accurate way.

The sentence has been deleted

Figure 1:

The authors may want to introduce some more quantitative ways (such as count of dark pixels) to support the arguments of "only 4Rtau is expressed in WT mice (Line 323)" and "4Rtau is negative in tWT transgenic mice (Line 327)". Reading from the figure in the PDF file, I can barely tell the difference between 4Rtau WT CA1 v.s. mtWT CA1 (row C, 1st and 2nd column), and find it hard to conclude that 4Rtau is expressed in WT mice while not in mtWT.

New figures representative of 4Rtau immunoreactivity in WT are included in Figure 1

Densitometry to 3Rtau and 4Rtau immunoreactivity in the CA1 region and DG has been assessed in the three groups of non-inoculated mice. The graphs are added to Figure 1.

Added text in Material and methods: 

Photomicrographs of sections of the dentate gyrus (DG) and CA1 stained with anti-3Rau and anti-4Rtau antibodies were obtained at a magnification of ×200, covering an area of 0.126 mm2, using a DP25 camera adapted to an Olympus BX50 light microscope. The pictures, two areas per region per case in every case, were analyzed using Photoshop software. Non-inoculated WT, mtWT, and hTau mice, and AD-inoculated hTau mice, surviving six months after injection, were assessed. The density of tau staining was calculated as the intensity of the diaminobenzidine (DAB) precipitate pigment normalized for the total area and expressed as a percentage of arbitrary units per area. Kolmogorov–Smirnov test was used to test the normality of the distribution. Results were analyzed with one‐way ANOVA and post hoc Tukey. Differences between groups were considered statistically significant at ***P < 0.001 when comparing WT, and ### P < 0.001 when comparing to mtWT.

Added text in Results

Densitometric studies further stressed the differences in the expression of 3Rtau and 4Rtau in the CA1 region ad dentate gyrus in the three groups of mice (Figure 1 graphs). As expected 4Rtau is the predominant form in WT mice, but the presence of 3Rtau-immunoreactive neurons in the inner region of the DG is also represented by a 3Rtau signal in the DG of WT mice. In contrast, 3Rtau is more abundant in the CA1 region and DG in hTau. mtWT mice do not show 3Rtau and 4Rtau expression.

Added text in the legend of Figure 1:

Graphs show quantitative densitometry for anti-3Rtau and anti-4Rtau immunostaining expressed as a percentage of arbitrary units per area in CA1 and dentate gyrus (DG) regions in WT, mtWT, and hTau mice. Results were analyzed with one‐way ANOVA and post hoc Tukey. Differences between groups were considered statistically significant at ***P < 0.001 when comparing hTau with WT, and ### P < 0.001 when comparing htau with mtWT. Note that 4Rtau is predominant in WT mice, excepting the DG in which 3Rtau positive neurons are also localized in the inner layer. In contrast 3Rtau predominates in CA1 and DG in hTau transgenic mice.

Figure 5:

Similar to Figure 1. The author claimed on Line 451 that "Deposits contained 4Rtau (Figure 5B, E), but 3Rtau was more abundant (Figure 5C and F)." However, 5C are visually similar to 5B, and 5F similar to 5E. It is therefore difficult to come to the same conclusion just by looking at the figures. Some quantitative measures will be helpful.

Changes added in Results

Curiously, 3Rtau and 4Rtau deposits were localized in the cytoplasm of neurons which was in contrast with the predominant localization of 3Rtau in the neuropil in non-inoculated hTau mice (see Figure 1 for comparison).

Differences in the intensity of the densitometric signal between 3Rtau and 4Rtau in the CA1 and DG of inoculated hTau mice are shown in the graph of Figure 5. 3Rtau-immunoreactivity is significantly higher than 4Rtau-immunoreactivity in the CA1 region. This pattern was not reproduced in the DG due to individual variations. 

Changes added in the legend of  Figure 5

Graph shows quantitative densitometry for anti-3Rtau and anti-4Rtau staining expressed as a percentage of arbitrary units per area in CA1 and dentate gyrus (DG) regions in inoculated hTau mice surviving six months after injection. Results were analyzed with Student’s t-test. Differences between anti-3Rtau and anti-4Rtau staining within the same region in consecutive slices were considered statistically significant at *P < 0.05.

Figure 2:

I had a hard time relating the bar plot with the gel image. In the bar plot, the 'WT' white bars across different x labels look almost of identical height. Did the authors scale the values of 'WT' to be 1? If so the authors may want to consider add bounding boxes between different x labels, or separate them into subplots, to prevent readers from comparing bars in one x label with bars in another.

Is the last lane of figure 2 the MW marker? Is the last lane label "hTau" correct? For 4Rtau, why is there two bands in the last lane but three arrows of MW?

Western blots were carried in parallel, and the molecular weights are approximate as indicated in the text.

This sentence is added in the corresponding figure legend

Since the samples were run in parallel the molecular weights in the right are also valid for the lanes of WT and mtWT homogenates

Two marks are now shown in the wb to label 4Rtau in hTau mice

Figure 3A: The authors may want to add the MW marker lane.

Added

Figure 3B:

The authors may want to use a table instead of a Venn diagram.

The font size of the text in 3B is too small to read.

The two circles take up a lot of empty space while its diameter or area seems not relevant to the number of phosphorylation sites. The blue circle has 19+8=27 sites, the yellow circle has 8+2=10 sites, but the two have the same size.

The Venn diagram has been substituted by a Table as suggested

Figure 4:

The authors may want to explicitly label the length of the scale bar instead of appending them at the figure caption. Doing so can greatly help readers to identify relevant objects in the figures.

The length of the bar is explicitly labelled in every figure

Line 499 and Figure 6:

It is unclear how the authors draw the brain diagrams. Are they reploted from the literature? If so citation is recommended.

It is unclear how the red square marks in Figure 6 were decided. Is Figure 6 a replot of both Figure 5 and Figure 4?

There seem to be some labels of the left brain in every figure of the Figure 6 but they are unreadable.

There are missing labels between the two columns. Are they independent repeats?

The 2nd row "mtWT inoc AD" and the 4th row "WT and hTau inoc control" look identical. If there is no any other information to read from the above 2nd and the 4th row except that "there is no deposit", the authors may want to replace the 2nd and the 4th rows with text "Deposit not detected".

The text of the legend to Figure 6

The distribution of abnormal tau deposits in WT, mtWT, and hTau unilaterally inoculated in the hippocampus with the same inocula of sarkosyl-insoluble fraction of AD and WT, and hTau inoculated with sarkosyl-insoluble fractions from one control brain. Mice were inoculated at the age of six months and killed at the age of nine months. Red points are tau deposits representative of deposits depicted in Figures 4 and 5. Tau deposits are not detected in mtWT inoculated with AD sarkosyl-insoluble fractions. In contrast, Tau deposits are found in WT and hTau inoculated with AD sarkosyl-insoluble fractions. Tau deposits in CA1 region and dentate gyrus are more abundant in inoculated hTau mice when compared with WT mice. Tau deposits in the stratum radiatum and stratum oriens are also more abundant in hTau than in WT mice. In contrast, tau deposits in the corpus callosum and fimbria (involving glial cells) are more copious in inoculated WT mice. No abnormal tau deposits are detected in WT and hTau mice inoculated with sarkosyl-insoluble fractions from a control case. The maps were obtained from the Atlas of Paxinos and Franklin [31] and represent two different coronal levels to facilitate the extent of tau deposits considering the three axis; sections of the left column are more anterior than the sections of the right column.    

Line 521, Figure 7:

The authors claimed "In contrast, cytoplasmic SRPK2-Ser497 immunoreactivity is found equally in neurons with and without phospho-tau deposits in hTau mice (D). " Is Figure 7D WITH or WITHOUT phospho-tau deposits?

Added in the text:

Cytoplasmic SRPK2-Ser497 immunoreactivity is found in neurons with phospho-tau deposits in inoculated hTau mice

Typo:

Line 327: 'tWT' -> 'mtWT'.

Table 2 header: 'inoc AD hau' -> 'inoc AD tau'

Line 480: Are the question marks typos?

Corrected

Round 2

Reviewer 2 Report

Acepted